# Research Progress on Fungal Sesterterpenoids Biosynthesis

**DOI:** 10.3390/jof8101080

**Published:** 2022-10-14

**Authors:** Ping Zhang, Jianzhao Qi, Yingce Duan, Jin-ming Gao, Chengwei Liu

**Affiliations:** 1Key Laboratory for Enzyme and Enzyme-like Material Engineering of Heilongjiang, College of Life Science, Northeast Forestry University, Harbin 150040, China; 2Shaanxi Key Laboratory of Natural Products & Chemical Biology, College of Chemistry & Pharmacy, Northwest A&F University, Yangling 712100, China

**Keywords:** sesterterpenes, sesterterpene synthase, terpene biosynthesis, cytochrome P450

## Abstract

Sesterterpenes are 25-carbon terpenoids formed by the cyclization of dimethyl allyl diphosphate (DMAPP) and isopentenyl diphosphate (IPP) as structural units by sesterterpenes synthases. Some (not all) sesterterpenoids are modified by cytochrome P450s (CYP450s), resulting in more intricate structures. These compounds have significant physiological activities and pharmacological effects in anti-inflammatory, antibacterial, antitumour, and hypolipidemic communities. Despite being a rare class of terpenoids, sesterterpenoids derived from fungi show a wide range of structural variations. The discovered fungal sesterterpenoid synthases are composed of C-terminal prenyltransferase (PT) and N-terminal terpene synthase (TS) domains, which were given the name PTTSs. PTTSs have the capacities to catalyze chain lengthening and cyclization concurrently. This review summarizes all 52 fungal PTTSs synthases and their biosynthetic pathways involving 100 sesterterpenoids since the discovery of the first PTTSs synthase from fungi in 2013.

## 1. Introduction

Over 80,000 terpene natural products have been identified in nature [1], making them one of the most abundant and structurally diverse natural product families. Terpenoids are synthesized from the universal C5 precursors, DMAPP and IPP. DMAPP and IPP can be generated by the mevalonate (MVA) or 2-C-methyl-D-erythritol-4-phosphate (MEP) pathways, depending on the species. Subsequently, PT assembles IPP and DMAPP into varying linear isoprenoid diphosphate chains including geranyl-farnesyl pyrophosphate (GFPP). Terpene synthase (TPS) generates these chains into terpene backbones with multiple chiral centers. Various enzymes modify the core backbone to produce structurally and functionally distinct terpenoids. Terpenoids are further classified according to the number of isoprene units employed in their formation as monoterpenes (C10), sesquiterpenes (C15), diterpenes (C20), sesterterpenes (C25), and triterpenes (C30). Compared to monoterpenes, sesquiterpenes, diterpenes, and triterpenes, sesterterpenes are the rarest terpenoids found to date, with approximately 1000 diterpenes isolated [2], mainly from fungi, bacteria, lichens, higher plants, insects, and various marine invertebrates [3].

Fungi-derived sesterterpenoids exhibit a diverse set of physiological and pharmacological properties. Fusaproliferin (**1**), discovered from the eggplant-disease fungus *Fusarium solani*, demonstrated powerful and rapid cytotoxic effects on both pancreatic and breast cancer cells, paving the way for the creation of more potent anti-pancreatic cancer medicines [4]. The metabolites of *Aspergillus flucculosus*, 14,15-dehydro-6-epi-ophiobolin K (**2**), 14,15-dehydro-ophiobolin K (**3**), 14,15-dehydro-6-epi-ophiobolin G (**4**), 14,15-dehydro-ophiobolin G (**5**), and 14,15-dehydro-(*Z*)-14-ophiobolin G (**6**) were extremely cytotoxic against six cancer cell lines, HCT-15, NUGC-3, NCI-H23, ACHN, PC-3, and MDA-MB-231, with GI_50_ values ranging from 0.14 to 2.01 µM [5]. The antibacterial effects of ophiobolin derivatives 3-anhydro-6-hydroxy-ophiobolin A (**7**) and Ophiobolin T (**8**) against *Bacillus subtilis*, Bacille *Calmette–Guerin* strain, *Staphylococcus aureus*, and *Methicillin-resistant S. aureus* are promising molecules, and they are expected to be the lead compounds for the development of new bacteriostatic agents [6,7]. Bipolarin E (**9**), derived from the phytopathogenic fungus *Bipolaris* sp. TJ403-B1, showed significant antibacterial action against *Enterococcus faecalis* and *Pseudomonas aeruginosa* [8]. The ophiobolin-type sesterterpenoids, Asperophobolins H–J (**10**–**12**) and 3-anhydro-6-epi-ophiobolin A (**13**), isolated from *Aspergillus* sp., showed significant inhibition of lipopolysaccharide-induced nitric oxide production by macrophages, and hence, had excellent anti-inflammatory activity [9]. Further investigations revealed that **12** substantially suppressed the macrophage production of IL-1, RANTES, MIP-1, and TNF-, and stimulated the release of IL-13 via blocking HO-1 induction and the NF-B pathway, providing a scientific basis for anti-inflammatory treatment [10]. Compound **12** [10] and Bipolarolides A (**14**) [11], isolated from the fungus *Bipolaris* sp. TJ403-B1, had a strong inhibitory effect on HMG-CoA reductase (Figure 1). Compound **14** was also found to have a hypolipidemic effect on HepG2 cells, with an effect comparable to that of the cholesterol-lowering drug pravastatin. As a result, compound **14** is a potential HMGR antagonist that merits further investigation [11]. These discoveries could pave the way for the development of novel HMG-CoA reductase inhibitors and anti-hyperlipidaemic drugs.

The results of sesterterpenes biosynthesis research from various species indicate that the PT domain and the TS domain are required functional modules for sesterterpene synthesis, and they, together, constitute bifunctional chimeric sesterterpene synthases. In plants, the two functional domains of PT and TS that comprise sesterterpene synthase are encoded by two genes independently [12], such as *AtGFPPS2-AtTPS18* from *Arabidopsis thaliana* (Figure 2A) [13]. In a few plants, the enzymes responsible for the synthesis of sesterterpenes harbor two functional domains, PT and TS domains, which are bifunctional chimeric enzymes, such as *LcTPS2* from *Leucoseptrum Canum* [14] (Figure 2B). Sesterterpenes are produced in bacteria by two distinct genes, each with a PT domain and TS domain, such as GFPPS-SmTS1 from *Streptomyces mobaraensis* (Figure 2C)—the former is responsible for the formation of GFPP, and the latter for the cyclization of GFPP to form a sesterterpene backbone [15]. In contrast, sesterterpene synthases in fungi are all chimeric enzymes with PT and TS functional domains, such as AcOS from *A. clavatus* [16] (Figure 2D).

With the advancement of sequencing technology in recent years, more and more genes with biological functions have been discovered. To date, 52 PTTSs (Table 1) from fungi have been identified, and their functions have been functionally validated via heterologous expression in *Escherichia coli*, *Saccharomyces cerevisiae*, or *A. oryzae*. Parts of the backbone generated by 52 PTTSs were further modified by other post-modifying enzymes on their gene cluster, yielding a total of 100 sesterterpenoids. This review provides a detailed summary of these enzymes and their biosynthetic pathways.

## 2. Characteristics and Classification of Fungal PTTSs

Fungal PTTSs are bifunctional enzymes with two domains, PT and TS. PT is responsible for catalyzing the combination of IPP and DMAPP to generate C25 GFPP, while TS catalyzes the cyclization of GFPP to generate various sesterterpenes. Phylogenetic analysis revealed that fungal diterpene synthases could be divided into two broad categories, clades I and II, which catalyze the formation of various ring systems according to their cyclization patterns [17].

### 2.1. Characteristics of Fungal PTTSs

Terpene synthases are typically classified into two classes, class I and class II, based on their initial carbon cation formation strategy. The former catalyzes an olefin cation cyclization reaction initiated by diphosphate (PPI) cleavage, while the latter catalyzes an olefin protonation reaction [18]. The N-terminus of class I terpene synthase sequences typically contain DDxxD and NSE/DTE-conserved motifs (Figure 3A), while the C-terminus contains two aspartic acid-rich-conserved regions, DDXXD and DDXXN (Figure 3A). A class II cyclase uses aspartic acid from the DXDD motif as a catalytic acid to protonate the polyisoprenoid’s terminal olefinic bond, resulting in a tertiary carbocation [19].

Sequence alignment revealed that all identified fungal PTTSs had the N-terminal motifs DDxxD and NSE/DTE, as well as the C-terminal motifs DDXXD and DDXXN, which are common features of class I terpene synthases [20], and on this basis, it is assumed that all known fungal PTTSs are class I terpene synthases. Analysis of the cyclization mechanism for sesterterpenes shows that the binding of three metal ions (frequency, Mg^2+^ > Mn^2+^ > Co^2+^) to GFPP induces a conformational change in GFPP, resulting in the closure of the enzyme active site and ionization of GFPP, which becomes an allylic cation and releases PPI [18]. Three-dimensional structural AcOS, the first sesterterpene synthase identified in a fungus, modeled by the trRosetta website (http://yanglab.nankai.edu.cn/trRosetta/, (accessed on 1 September 2022)), revealed two distinct structural domains, with sequence analysis indicating that the TS structural domain is at the N-terminal, while the PT structural domain is located at the C-terminal (Figure 3B).

### 2.2. Phylogenetic Analysis of Fungal PTTSs

A phylogenetic analysis of 52 fungal-derived PTTSs revealed six distinct groups (clusters A–F). According to the cyclization pattern, clusters A, E, and F belong to clade I, while clusters B–D belong to clade II (Figure 4).

### 2.3. Catalytic Characteristics of Fungal PTTSs

Clades I and II, respectively, catalyze type A (C1-IV-V) and type B (C1-III-IV) cyclization [21]. All TS domains in clusters A, F, and E initiate cyclization by forming a 15-5-membered ring system [22]. The type A ring system is formed by the sequential cyclization reaction of the C1 cation, the C14-C15 alkene (IV), and the C18-C19 alkene (V) of GFPP (C1-IV-V). In the initial cyclization step, all known TS structural domains in clusters B, C, and D produce an 11-5-membered ring system [23]. The cyclization reaction between the C1 cation, the C10-C11 alkene (III), and the C14-C15 alkene (IV) of GFPP produces the B-type ring system (C1-III-IV). The structures of compounds from 52 fungal PTTSs and their possible cyclizations are shown in Figure 5.

## 3. Fungal Sesterterpene Biosynthesis Pathway

The biosynthetic pathways of sesterterpenes in fungi are all confirmed by a heterologous expression, with *S. cerevisiae*, *A. oryzae*, and *E. coli* being common hosts. In fungi, the majority of sesterterpenes are synthesized by single PTTSs with no post-modification. A small number of sesterterpenes on the gene cluster modifies the final compound via CYP450s. There are also very few functional genes encoding flavin-dependent oxidases on the biosynthetic gene clusters of sesterterpenes.

### 3.1. Sesterterpene Yielded by a Single PTTS

The first PTTS, AcOS, was discovered serendipitously by the Hideaki Oikawa lab in 2013 during the identification of possible diterpene synthase coding genes in public databases using the *A. oryzae* expression system [16]. The *A. oryzae* transformant of *AcOS* produced four sesterterpenoids, but in the enzymatic reaction in vitro, *AcOS* produced diterpene in addition to sesterterpenes. These four sesterterpenoids include Ophiobolin F (**15**), Ophiobane 1 (**16**), Ophiobolane 2 (**17**), and Clavaphyllene (**18**). Of these four metabolites, **15**–**17** are 5/8/5-fused tricyclic sesterterpenes [16]. AcOS contains two domains, the N-terminal PT domain and the C-terminal GFPP synthase domain, which are identical to diterpene synthases, according to the protein sequence analysis of the cDNA expressed in *E. coli* [16]. With the discovery of the first PTTS, the pathways of sesterterpenes derived from plants, fungi, and bacteria have been gradually identified.

Sequence analysis indicated that the C-terminus of AcOS contains the αα domain responsible for the generation of linear GFPP, while the N-terminus contains the cyclase domain responsible for the formation of Ophobolin F (**15**) [24]. PaFS, a diterpene synthase found in the phytopathogenic fungus *Phomopsis amygdali* [25], is 41% identical with AcOS. They both catalyze the cyclization reactions of geranylgeranyl pyrophosphate (GGPP) and GFPP via a common 5–11 ring system intermediate. Sequence alignment showed that the majority of residues in PaFS and AcOS are conserved, but two pairs of residues, W225 and V228 in PaFS and L217 and A220 in AcOS, differ significantly in size [24]. The α-domain at C-terminal of PaFS, which is responsible for GGPP synthase, was swapped with the α-domain of EvSS, which is responsible for GFPP synthesis, but no sesterterpenes were produced. Therefore, it is speculated that the active pocket of the PaFS cyclase domain is too small to accommodate GFPP [24].

The N-terminal cyclase domain of the chimera PaFS underwent a double mutation (W225L/V228A), yielding three unknown sesterterpene products. The chimera obtained by replacing the GFPP synthase domain on the C-terminal of EvSS with the GGPP synthase domain of PaFS produced only diterpenes and sesquiterpenes. A double mutant L217W/A220V was made on the N-terminal cyclase structural domain of this chimera AcOS, which produced not sesterterpenes, but diterpenes and sesquiterpenes [24]. Thus, the substitution of crucial amino acid residues can change the size of the active packet, resulting in the formation of the corresponding compounds.

Chai et al. conducted a systematic study on the biosynthesis of ophiobolin in *A. ustus* in 2016 [26]. Au8003 was confirmed to be the synthase responsible for ophiobolin F (**15**) in *A. ustus* by gene deletion and complementation experiments [26]. Subsequently, Yuan et al. attempted to increase the yield of ophiobolin F (**15**) by optimizing the codons of synthase, adding substrates, and optimizing the pathway of precursor supply [27].

Qin et al. and Matsuda et al. reported two bifunctional chimeric terpene synthases, EvVS [28] and EvSS [29] from *Emericella variecolor* in 2015. The heterologous expression of EvVS in *A. oryzae* only produced the diterpene varidiene, whereas the in vitro reaction of EvVS produced not only varidiene, but also sesterterpene (2*E*)-α-Cericene (**19**). The TS domain of EvVS was combined with the PT domain of EvSS to create a chimeric enzyme that was expressed in *A. oryzae* to produce **19** and diterpene varidiene [28]. Matsuda et al. con- 195 tinued to mine the genome of *E. variecolor* in 2016 and discovered another PTTS, EvAS, 196 and its heterologous expression in *A. oryzae* produced a new sesterterpene astellifadiene 197 (**20**) with a fused 6-8-6-5 ring [30].

Narita et al. discovered four sesterterpene synthase genes from phytopathogenic fungi in 2017, including BmTS1, BmTS2, and BmTS3 from *B. maydis*, as well as PbTS1 from *P. betae*. The four sesterterpenes Bm1-3 (**21**–**23**) and Pb1 (**24**) were produced by heterologous expression of these four genes in *A. oryzae* and *E. coli*, respectively [31].

In 2017, Bian et al. discovered two new sesterterpenes, mangicdiene (**25**) and variecoltetraene (**26**), via the heterologous expression of FgMS from *F. graminearum* using an optimized terpene substrate supplying *S. cerevisiae* as a chassis. The active sites F65L and F159G on FgMS were then mutated, yielding compound **27** and five possible sesterterpenes [32].

A survey of 34 possible PTTS-encoding genes derived from Ascomycete identified 28 PTTSs and 15 sesterterpenes, and these PTTSs were characterized via high-throughput automated platforms and efficient yeast chassis in 2021 [33]. These enzymes and compounds included Sesterfisherol (**28**) from AaSS; Bm3 (**23**) from MpBS and CfBS; Pb1 (**24**) from ChPS, CoFS, CiGS, and CsPS; sesterorbiculene (**29**) from CiSS, CoSS, CgSS; fusaproliferene (**30**) from CoFS; (-)-variculatriene B (**31**) from LmVS, PoVS1, ChVS, CsVS, and PoVS2, variculatriene A (**32**); clavaphyllene (**18**) from PfVS; ophiobolin F (**15**) from AuOS, and BmOS; preasperterpenoid A (**33**) from TvPS; sesterbrasiliatriene (**34**) from PaSS, brassiteraene A (**35**), brassiteraene B (**36**) from ChBS; sesterevisene (**37**) from ZbSS; β-geranylfarnesene (**38**) from CiGS, and PfVS; geranylfarnesol (**39**) from AnGS, PgGS, TaGS, and AaGS. Among the above compounds, sesterevisene (**37**) and sesterorbiculene (**29**) are two new cyclic sesterterpenoids [33].

In 2022, Jiang et al. discovered two novel PTTSs, CsSS and NnNS, from *Cytospora schulzeri* 12, 565; and *Nectria nigrescens* 12, 199, respectively, and identified their functions through heterologous expression in *A. oryzae* and *S. cerevisiae*. CsSS generates 5/12/5 tricyclic A-type sesterterpene schultriene (**40**), while NnNS produces the first 5/11 bicyclic B-type sesterterpene nigtetraene (**41**) to date [34].

**Table 1 jof-08-01080-t001:** The identified PTTSs from fungi and their products.

Entry	Genebank No./Gene	Producer	Host/Chassis	Product	Reference
AcOS	ACLA_76850	*Aspergillus clavatus*	*A.**oryzae*, *E. coli*	Ophiobolin F(**15**), Ophiobolane 1 (**16**), Ophiobolane 2 (**17**), Clavaphyllene (**18**)	[16]
Au8003	QIH97826.1	*Aspergillus ustus*	*E. coli*	Ophiobolin F(**15**)	[26]
BmOS	MW798226	*Bipolaris maydis*	*S. cerevisiae*	Ophiobolin F (**15**)	[33]
AuOS	MW798208	*Aspergillus ustus*	*S. cerevisiae*	Ophiobolin F (**17**)	[33]
PfVS	MW798216	*Pestalotiopsis fici*	*S. cerevisiae*	Clavaphyllene (**18**), Variculatriene A (**32**), β-Geranylfarnesene (**38**)	[33]
EvVS	LC063849	*Emericella variecolor*	*A. oryzae*, *E. coli*	(2E)-a-cericerene (**19**)	[28]
EvAS	LC113889	*Emericella varecolor* NBRC 32302	*A. oryzae*, *E. coli*	Astellifadiene (**20**)	[30]
BmTS1	EMD84919	*Bipolaris maydis* ATCC48331	*A. oryzae* *E. coli*	Bm1 (**21**)	[31]
BmTS2	EMD93209	*Bipolaris maydis* ATCC48331	*A. oryzae* *E. coli*	Bm2 (**22**)	[31]
BmTS3	EMD93704	*Bipolaris maydis* ATCC48331	*A. oryzae* *E. coli*	Bm3 (**23**)	[31]
CfBS	MW798209	*Colletotrichum fioriniae*	*S. cerevisiae*	Bm3 (**23**)	[33]
MpBS	MW798229	*Macrophomina phaseolina*	*S. cerevisiae*	Bm3 (**23**)	[33]
PbTS1	LC274619	*Phoma betae* PS-13	*A. oryzae* *E. coli*	Pb1 (**24**)	[31]
BtcA_Co_	N4V6D4.1	*Colletotrichum orbulare*	*A. oryzae*	Pb1 (**24**)	[35]
ChPS	MW798213	*Colletotrichum higginsianum*	*S. cerevisiae*	Pb1 (**24**)	[33]
CsPS	MW798219	*Colletotrichum siamense*	*S. cerevisiae*	Pb1 (**24**)	[33]
CoFS	MW798210	*Colletotrichum orbiculare*	*S. cerevisiae*	Pb1 (**24**), Fusaproliferene (**30**)	[33]
CiGS	MW798200	*Colletotrichum incanum*	*S. cerevisiae*	Pb1 (**24**), β-Geranylfarnesene (**38**)	[33]
FgMS	AQY56777	*Fusarium graminearum*	*S. cerevisiae*, *A. oryzae*, *E. coli*	Mangicdiene (**25**), Variecoltetraene (**26**)	[32,36]
NfSS	EAW16201	*Neosartorya fischeri*	*A. oryzae*, *E. coli*	Sesterfisherol (**28**)	[22]
AaSS	MW798204	*Alternaria alternata*	*S. cerevisiae*	Sesterfisherol (**28**)	[33]
CiSS	MW798201	*Colletotrichum incanum*	*S. cerevisiae*	Sesterorbiculene (**29**)	[33]
CoSS	MW798211	*Colletotrichum orbiculare*	*S. cerevisiae*	Sesterorbiculene (**29**)	[33]
CgSS	MW798218	*Colletotrichum gloeosporioides*	*S. cerevisiae*	Sesterorbiculene (**29**)	[33]
ChVS	MW798212	*Colletotrichum higginsianum*	*S. cerevisiae*	(-)-Variculatriene B (**31**)	[33]
PoVS	MW798215	*Pyricularia oryzae*	*S. cerevisiae*	(-)-Variculatriene B (**31**)	[33]
LmVS	MW798221	*Lophiostoma macrostomum*	*S. cerevisiae*	(-)-Variculatriene B (**31**)	[33]
CsVS	MW798223	*Colletotrichum sublineola*	*S. cerevisiae*	(-)-Variculatriene B (**31**)	[33]
PoVS	MW798227	*Pyricularia oryzae*	*S. cerevisiae*	(-)-Variculatriene B (**31**)	[33]
TtPS	MW798214	*Thermothielavioides terrestris*	*S. cerevisiae*	Preasperterpenoid A (**33**)	[33]
PvPS	LC228602	*Penicillium verruculosum*	*A. oryzae*, *E. coli*	Preasperterpenoid A(**33**)	[37]
AsTC	MK140602	*Talaromyces wortmannii*	*A. oryzae*, *E. coli*	Preasperterpenoid A(**33**)	[38]
TvPS	MW798225	*Talaromyces verruculosus*	*S. cerevisiae*	Preasperterpenoid A (**33**)	[33]
PaSS	MW798222	*Penicillium arizonense*	*S. cerevisiae*	Sesterbrasiliatriene (**34**)	[33]
PbSS	LC228601	*Penicillium brasilianum*	*A. oryzae*, *E. coli*	Sesterbrasiliatriene (**34**)	[37]
ChBS	MW798232	*Colletotrichum higginsianum*	*S. cerevisiae*	Brassiteraene A (**35**), Brassiteraene B (**36**)	[33]
ZbSS	MW798202	*Zymoseptoria brevis*	*S. cerevisiae*	Sesterevisene (**37**)	[33]
AnGS	MW798203	*Aspergillus niger*	*S. cerevisiae*	Geranylfarnesol (**39**)	[33]
PgGS	MW798206	*Penicillium griseofulvum*	*S. cerevisiae*	Geranylfarnesol (**39**)	[33]
PgFS	MW798217	*Pyricularia grisea*	*S. cerevisiae*	Geranylfarnesol (**39**)	[33]
TaGS	MW798220	*Thielavia arenaria*	*S. cerevisiae*	Geranylfarnesol (**39**)	[33]
AaGS	MW798231	*Aspergillus aculeatus*	*S. cerevisiae*	Geranylfarnesol (**39**)	[33]
CsSS	MW685620	*Cytospora schulzeri* 12,565	*A. oryzae*, *E. coli*	Schultriene (**40**)	[34]
NnNS	MW685621	*Nectria nigrescens* 12,199	*A. oryzae*, *E. coli*	Nigtetraene (**41**)	[34]
EvSS	LC073704	*Emericella variecolor*	*A. oryzae*, *E. coli*	Stellata-2,6,19-triene (**42**)	[29]
EvQS	LC155210	*Emericella variecolor*	*A. oryzae*, *E. coli*	Quannulatene (**44**)	[39]
AcldAS	CEL06489.1	*Aspergillus calidoustus* CBS121601	*A. oryzae*, *E. coli*	Asperterpenol A (**47**)	[40]
AuAS	MW387950	*Aspergillus ustus* 094102	*A. oryzae*	Aspergildiene A (**49**) Aspergildiene (**50**–**53**)	[41]
FoFS	MW446505	*Fusarium oxysporum*	*A. oryzae*, *E. coli*	Fusoxypenes A-C (**58**–**60**), (-)-astellatene (**61**)	[42]
AtAS(StTA)	ATEG_03568 (KX449366)	*Aspergillus terreus*	*A. oryzae*	Preaspterpenacid I (**62**)	[42,43]
PsTA	NA	*Penicillium herquei* TJ403-A1	*A. oryzae*, *E. coli*	Penisentene (**64**)	[44]

NA indicates not available.

### 3.2. Sesterterpenoid Yielded by PTTS and CYP450

In addition to PTTSs, the biosynthetic gene cluster (BGC) of a small number of sesterterpenes contains CYP450, which is responsible for post-cyclization modifications. Matsuda et al. discovered the PTTS, Stl-SS (EvSS), in the *E**. variecolor* genome using AcOS as a probe in 2015, and expressed it heterologously in *A.*
*oryzae* NSAR1 to generate a sesterterpene stellata-2,6,19-triene (**42**) with an 11-5 tricyclic ring. The CYP450 monooxygenase Stl-P450 located near Stl-SS converts this compound to stellatic acid (**43**) (Figure 6A) [29].

Sato et al. discovered a PTTS NfSS from *Neosartorya fischeri* using a genome mining strategy in 2015. In *A. oryzae*, the heterologous expression of this gene resulted in sesterfisherol (**28**) with the 5-6-8-5 tetracyclic ring. CYP450 monooxygenase (NfP450) converts **28** to sesterfisheric acid (**46**) (Figure 6B). Compounds **28** and **46** have not been identified in *N. fischeri* metabolites, indicating that this BGC is silent. F191A, a site-directed mutagenesis of NFSS, produced novel sesterterpenes, but not **28**, confirming that phenylalanine F191 is a critical amino acid residue for the production of **28** [45].

The regioselectivity and stereoselectivity in terpene formation reactions are determined by the conformations of carbocation intermediates that mirror the initial conformation of GFPP [46]. By calculating the intrinsic atomic mobility of the carbon positive ion intermediate during the enzymatic catalysis of sesterterpenes (**28** and **44**), Sato et al. found that the two methyl groups (C20 and C23) remained unchanged during the first half of the cyclization cascade. It is also suggested that the **28** and **44** enzyme-catalyzed process is divided into three stages: (I) formation of the 5/12/5 tricycles, (II) conformational change and hydrogen transfer, and (III) ring rearrangement to form the 5-6-8-5 ring system [46].

With the help of genome mining, Quan et al. found another sesterterpene BGC in *A. calidoustus* in 2020. The cluster contains two genes: AcldAS, a PTTS; and AcldA-P450, a CYP450 family member. AcldAS was discovered through heterologous expression and in vitro enzymatic assays to produce the sesterterpene asperterpenol A (**47**), which was then oxidized by AcldA-P450 to asperterpenol B (**48**) (Figure 6C). Asperterpenol A have a unique 6/6/8/5 tetracyclic ring system, and isotope labeling was used to determine the absolute configuration and cyclization mechanism of **47** [40].

Guo et al. discovered a multi-product sesterterpene biosynthesis gene cluster in *A. ustus* genome in 2021. A bifunctional terpene synthase AuAS and a CYP450 monooxygenase AuAP450 were found in this BGC. By a heterologous expression in *A. oryzae*, AuAS produced five novel sesterterpenes, aspergiltriene A (**49**), and aspergildiene A–D (**50**–**53**). Based on this, the AuAP450-introduced *A. oryzae* producer synthesized four new sesterterpene alcohols, aspergilols A–D (**54**–**57**) (Figure 6D). Only **49** has a 5/12/5 tricyclic skeleton, whereas the other four have a 5/6/8/5 tetracyclic skeleton [41]. The compound **49** is thought to be an earlier by-product produced in this pathway.

Jiang et al. identified the fungal chimeric terpene synthases FoFS and AtAS from the plant pathogenic fungi *F. oxysporum* and *A. terreus* in 2021, respectively. AtAS is otherwise known as STtA, as also reported by Clevenger et al. in 2017 [43]. The heterologous expression of the former produces fusoxypenes A–C (**58**–**60**) and (-)-astellatene (**61**), and the heterologous expression of the latter produces preaspterpenacid I (**62**) [42]. The compounds **58** and **62** are enantiomeric sesterterpenes with a 5−6−7−3−5 ring system, catalyzed by FoFS and AtAS, respectively. The C22 of **62** is then modified oxidatively by CYP450 to generate preaspterpenacid II (**63**) [42] (Figure 6E). Furthermore, the density functional calculations of FoFS-catalyzed reactions show that the formation of the pentacyclic system is a highly organized process [42].

Qiao et al. established a BGC consisting of a PTTS-encoding gene *pstA* and a CYP450 gene *pstB* by the genome mining of *P. herquei* TJ403-A1. The heterologous expression of *pstA* in *A. oryzae* yielded penisentene (**64**), and its co-expression with *pstB* produced two novel penisentene derivatives, penisentenol (**65**) and penisentone (**66**) (Figure 6F). Notably, the compound **64** possesses a unique 5/15 *cis*-fused ring system; in contrast, all other ring systems generated by known PTTSs are in trans. Thus, PsTA is the first PTTS to catalyze sesterterpenoids with a 5/15 cis-fused ring system. This unprecedented mode of ring fusion suggests that PsTA controls the stereochemical product of the initial cyclization in a novel mechanism [44].

### 3.3. Sesterterpene Yielded by the BGC Containing Multiple Genes

The post-modifying enzymes in addition to the typical cytochrome P450 include flavin-dependent oxidases in a minimal number of fungal sesterterpene BGC. Additionally, MFS transporters can occasionally be found in gene clusters involved in sesterterpene biosynthesis.

AcOS was the first report of PTTS [16], and subsequent studies showed that AcOS formed a functional *obl* BGC in *A. clavatus*, with genes encoding oxidases and transporters clustered around the AcOS-encoding gene [47]. The *obl* BGC was also found in the genomes of two other filamentous fungi, *B. maydes* and *E. variecolor*, and not only in *A. clavatus*. In addition to TPPCs OblA_Ac/Bm/Ev_ and CYP450 monooxygenase (OblB_Bm/Ac/Ev_) in the BGC of ophiobolin compounds from the genome of *A. clavatus*, there are also flavin-dependent oxidases (OblD_Bm/Ac/Ev_) in the *obl* BGCs (Figure 7A) [47]. AcOS was, therefore, named OblA_AC_.

These ophiobolin F (**15**) synthases, OblA_Bm/Ac/Ev_ [47], AuOS [33], AU8003 [26], and BmOS [33], showed a high identity of 61% to 100%. The products of OblA_AC_ (AcOS) include **16**–**18**, in addition to **15.** The heterologous expression of *oblA_Bm/Ac/Ev_* in *A. oryzae* yielded ophiobolin F (**15**), and its co-expression with *oblB_Ac/Ev_* produced four oxidized Ophiobolin derivatives (**67**–**70**). However, the co-expression of *oblA_Ac_* with *oblB_Bm_* did not result in the production of new compounds. Then, the transporter gene *oblD_Bm_* was co-expressed with *oblA_Ac_* and *oblB_Bm_*, leading to the three new sesterterpenes—ophiobolin (**71**), 6-epiophiobolin C (**72**), and 6-epiophiobolin N(**73**) [47] (Figure 7B). Combining the co-expression of *oblC_Ac_* with *oblA_Ac_*, *oblB_Bm_*, and *oblD_Bm_* in *A. oryzae* led to compound **74**. The compound **75** from *B. maydes* is presumed to have been further oxidized to form **76** (Figure 7B). Multiple novel sesterterpenes were produced by this combinatorial expression strategy of genes from various origins, revealing novel information about the biosynthetic process for ophiobolins [47].

Ophiobolin A (**76**) is a toxic sesterterpene metabolite produced by the pathogen *Cochliobolus miyabeanus* and *B. maydis.* In *C. miyabeanus*, inactivating the *PPT1* gene resulted in a 10-fold increase in the production of **76**. This means that inhibiting polyketide or non-ribosomal peptide biosynthesis, or both, promotes the production of sesterterpene metabolites [48].

In 2017, Mitsuhashi et al. predicted two new PTTSs, PbSS and PVPs, from *P. brasilianum* and *P. verruculosum*. The heterologous expression of these two genes in *A. oryzae* produced sesterbrasilitriene (**34**) and preasperterpenoid A (**33**), respectively [37]. In 2019, Hang et al. discovered a sesterterpene BGC containing three genes from the genome of *Talaromyces wortmannii* (Figure 8A). The subsequent recombination of this BGC in *A. oryzae* revealed that AsTC encodes a sesterterpene cyclase that is also capable of synthesizing preasperterpenoid A (**33**). AsTC has a high degree of homology with TvPS, PvPS, and TtPS, with an identity of at least 37%. The CYP450 oxidase AsTB then multioxidizes compound **33** to produce a new asperterpenoid A (**77**) and byproduct asperterpenoid B (**78**). Compound **77** continued to be further oxidized by another CYP450 oxidase, *astA*, to a new sesterterpenoid, asperterpenoid C (**79**) (Figure 8B) [38].

Gao et al. discovered two silent BGCs involved in the biosynthesis of betaestacin I (**24**) in two plant pathogenic fungi, *P. betae* and *C. orbulare*, in 2018. Both BGCs contain a PTTS (BtcA_Pb_/BtcA_Co_) and two CYP450 oxidases (Figure 9A), and the heterologous expression of *btcA_Pb_* or *btcA_Co_* results in the production of betaestacin I (**24**). There are two pairs of homologous CYP450s in these two BGCs. The first pair of CYP450s, BtcB_Pb_ and BtcB_Co_, catalyzed multiple oxidations of **24** with varying degrees of oxidation. The compound **24** is further oxidized by *btcB_Pb_* to betaestacin II (**80**) in the Btc BGC pathway from *P. betae*, whereas **24** is oxidized by *btcB_Co_* to betaestacin III (**81**), and further oxidized to IV (**82**) in the *btc_Co_* pathway. The modifications of the second pair of homologous CYP450s, BtcC_Pb_ and BtcC_Co_, were different. BtcC_Pb_ was unable to oxidize **80** further, whereas *btcC_Co_* catalyzed the multistep oxidation of **82**, yielding four new sesterterpenoids, betaestacinVa-c (**83**–**85**) and VI (**86**) (Figure 9B) [35].

During genome mining of *B. maydis* in 2018, Narita et al. discovered a BGC consisting of four genes (Figure 10A). The heterologous expression of the bifunctional terpene synthase (TpCA) in *A. oryzae* produces preterpestacin I (**23**). The compound **23** is hydroxylated regioselectively by CYP450 (TpCB) to produce preterpestacin II (**87**), which is then further oxidized to the carboxyl group to form **88**. The two-step oxidation of **87** catalyzed by a second cytochrome P450 (TpCC) yields vicinal diol moiety on the α-ring to form preterpinomycin III (**89**). The oxidation of the two hydroxyl groups on the A-ring of **89** is catalyzed by flavin-dependent oxidase (TpCD) to form a diketone, which forms **90** via a keto-enol interchange [49].

FgMS is a PTTS from *F. graminearum* responsible for the production of **25** and was identified by Bian et al. in 2017 [32]. Yuan et al. conducted a deeper excavation of the genome to reveal a BGC (Figure 11A) containing FgMS and characterized the BGC with the help of the efficient *A. oryzae* chassis, showing that this BGC is responsible for the synthesis of mangicols. The co-expression of *mgcD* (FgMS) and *mgcCE* produced mangicols A–E and H–J (**91**–**98**), and mangicols K–L (**99**–**100**) were obtained by introducing *mgcF* into the transformant of *mgcCDE* (Figure 11B) [36]. MgcE is a multifunctional CYP450 capable of catalyzing hydroxylation, epoxidation, and carbonylation reactions at the isoprene tail (C17–C20) of mangicdiene, and is widely involved in the formation of **91**, **93**–**94**, **96**–**98** (Figure 11B). MgcF is another multifunctional P450 that catalyzes hydroxylation at C7 or C8, respectively (Figure 11B) [36]. Given the excellent anti-inflammatory activity of **98**, multiple copies of *tHMG1* and *mgcCE* were randomly inserted into the host chromosome to increase the yield to 12.09 mg/L, a 151-fold increase compared to the initial heterologous expression strain [36].

## 4. Discussion

A total of 52 PTTSs were identified from 2013–2022, and most of them were functionally validated by heterologous expression using *A. oryzae*, *E. coli*, or *S. cerevisiae*. The heterologous expression enables PTTSs to generate a large number of new sesterterpenes, many of which have novel structural backbones. In fact, there are also numerous biosynthetic pathways for sesterterpenes in fungi that have not yet been identified. Most of these compounds have significant physiological activities, such as **1** with an anti-cancer effect [4], **2**–**6** with cytotoxicity [5], **7** and **8** with antibacterial activity [6,7], **10**–**13** with anti-Parkinsonian and anti-inflammatory properties [9], as well as **14** with a cholesterol-lowering effect [11]. These sesterterpenoids have potential applications in clinical treatment.

In recent years, the ongoing advancement of metabolomics and genome mining technologies has made it much easier to find sesterterpene synthases. Additional oxidative modifications such as cytochrome P450 provide further structural diversity of natural sesterterpenes. The rapid advancement of sesterterpene synthase research will be facilitated by the high-efficiency *A. oryzae* heterologous expression system for the targeted insertion of foreign genes into high expression sites [50], and the high-throughput automated protoplast transformation platform [36]. Although the chemical synthesis of some sesterterpenes with distinctive structures and intricate frameworks is challenging, heterologous biosynthesis using microorganisms as the chassis can produce target molecules quickly and effectively. The development of natural product chemistry will also be influenced by synthetic biology, which employs biosynthesis to create highly valuable sesterterpenes in medicine, health, and food chemistry.

## Figures and Tables

**Figure 1 jof-08-01080-f001:**
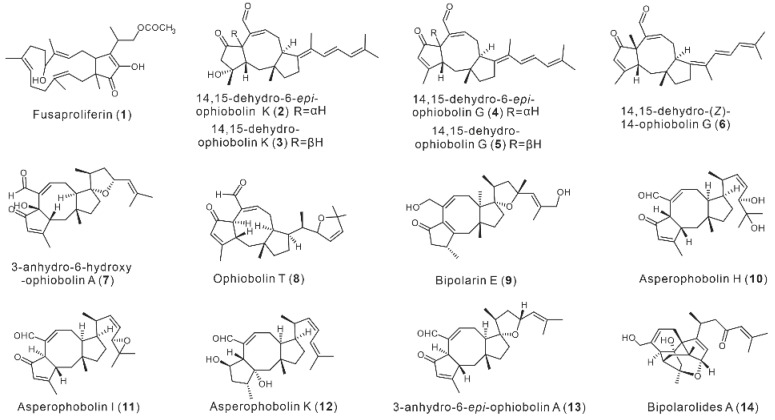
Representative natural sesterterpenes of fungi.

**Figure 2 jof-08-01080-f002:**
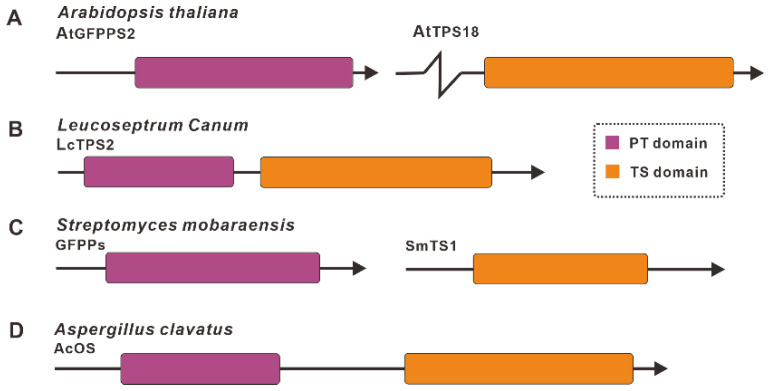
Schematic diagram of functional domains (PT and TS) associated with the biosynthesis of sesterterpene from plants (**A**,**B**), bacteria (**C**), and fungi (**D**). The prediction of functional domains was performed by PFAM (https://pfam.xfam.org (accessed on 1 September 2022)).

**Figure 3 jof-08-01080-f003:**
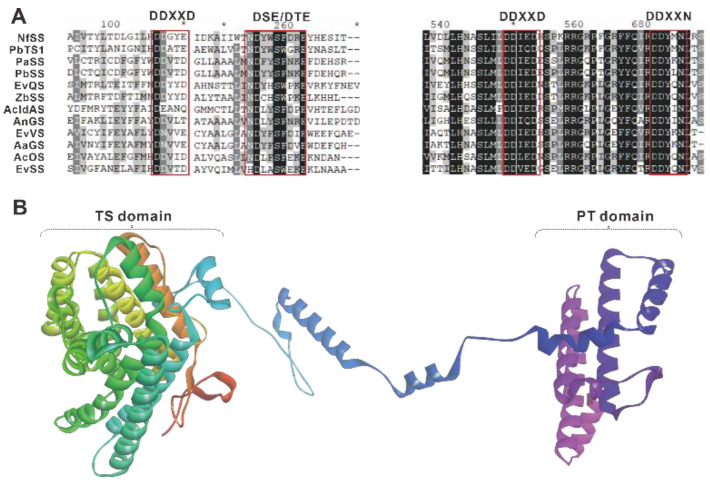
Motif analysis of fungal PTTSs (**A**) and domain analysis of AcOS (**B**). The DNAMAN package was used to compare 12 protein sequences of characteristic PTTSs from six clusters, whose GenBank accession no. are shown in Table 1. “*” indicates postions which have a single, fully conserved residue.

**Figure 4 jof-08-01080-f004:**
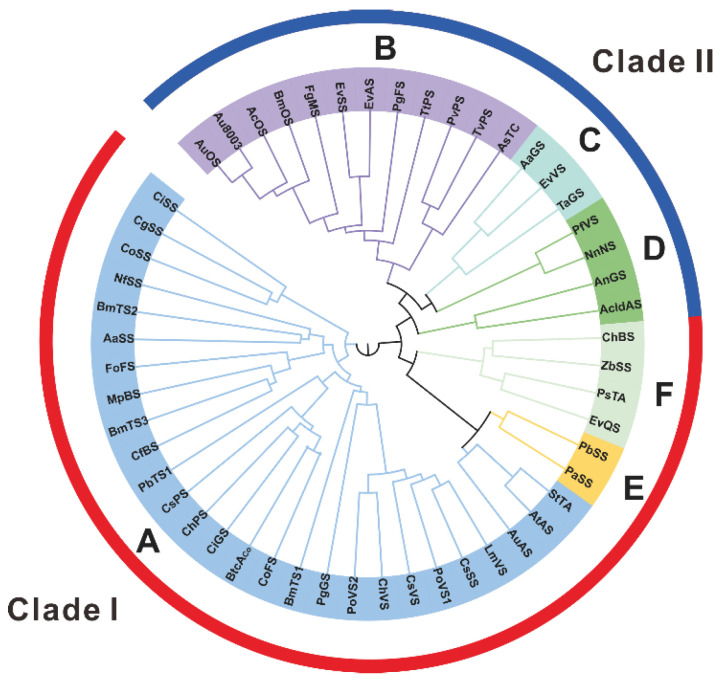
Phylogenetic tree analysis of fungal PTTSs with the maximum-likelihood method. The phylogenetic tree was constructed by IQ-TREE v 1.6.9 based on the full-length sequences of 52 PTTSs and presented in Figtree v1.4.4. A–F denote six subfamilies of PTTSs, where A, E, and F are in clade I, and B, C, and D are in clade II.

**Figure 5 jof-08-01080-f005:**
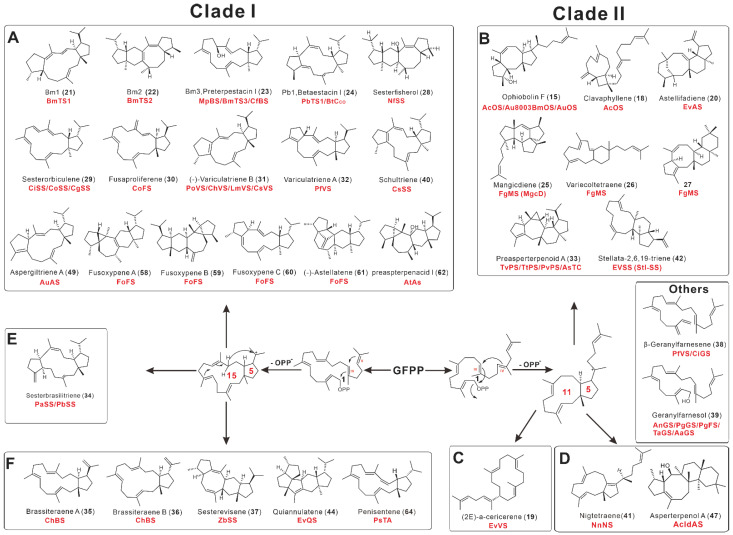
Summary of cyclization patterns of 52 fungal-derived PTTSs. (**A**–**F**) correspond to the six subfamilies of PTTSs.

**Figure 6 jof-08-01080-f006:**
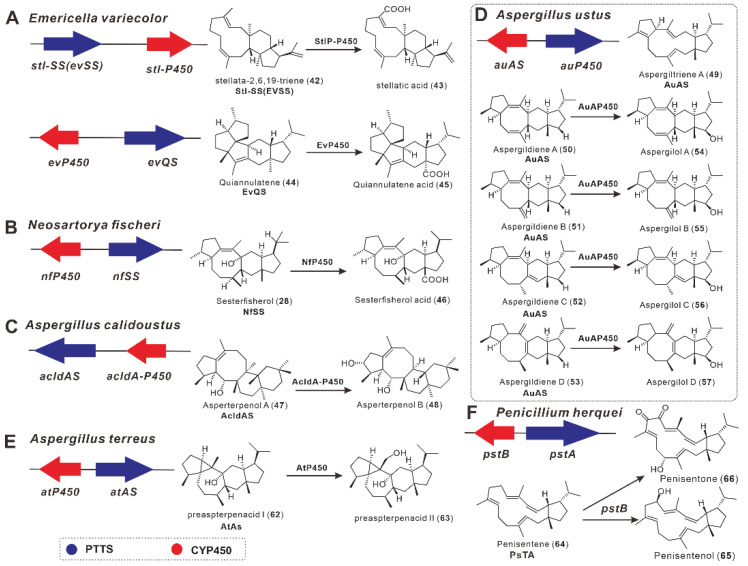
The BGCs composed of PTTSs and CYP450s and their biosynthesis. (**A**–**F**) represent six fungal sources of BGCs composed of PTTSs and CYP450s.Okada et al. discovered another PTTS, EvQS, from *E. variecolor* through genome mining in 2016, and expressed it in *A. oryzae* to produce quannulatene (**44**). Compound **44** was then oxidized to quannulati acid (**45**) by CYP450 enzymes located near the EvQS gene (A). Feeding experiments with isotopically labeled acetates and in vitro experiments with GFPP isotopoisomer revealed that **44** cyclizes in an unprecedented manner to form a unique fused 5-6-5-5-5 ring [39].

**Figure 7 jof-08-01080-f007:**
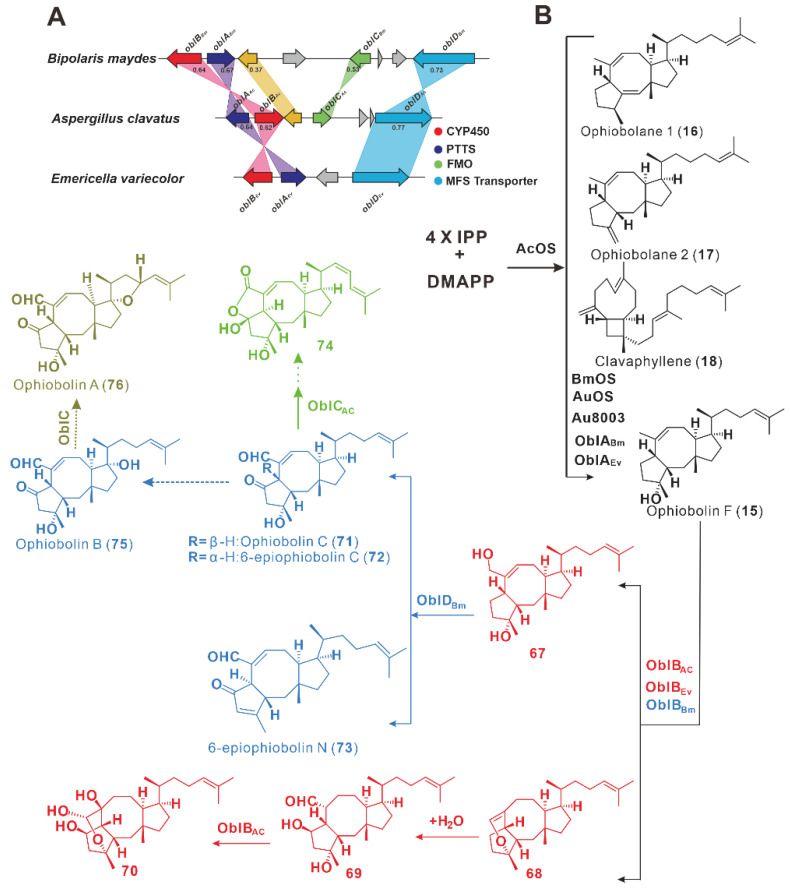
The *obl* BGCs containing multiple genes (**A**) and their biosynthesis (**B**).

**Figure 8 jof-08-01080-f008:**
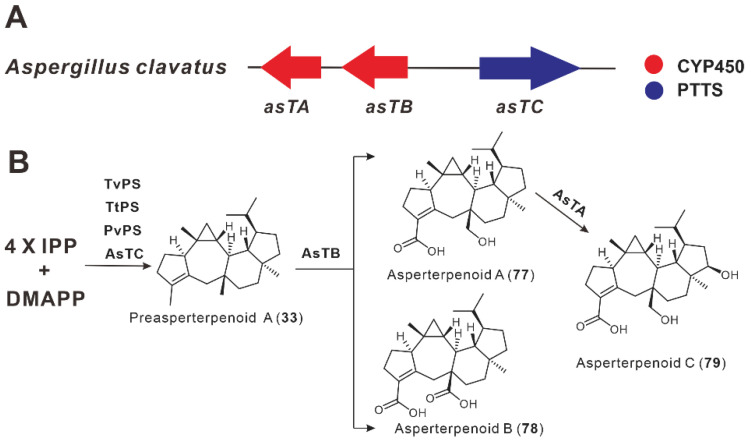
The BGC for asperterpenoids (**A**) and its biosynthetic pathway (**B**).

**Figure 9 jof-08-01080-f009:**
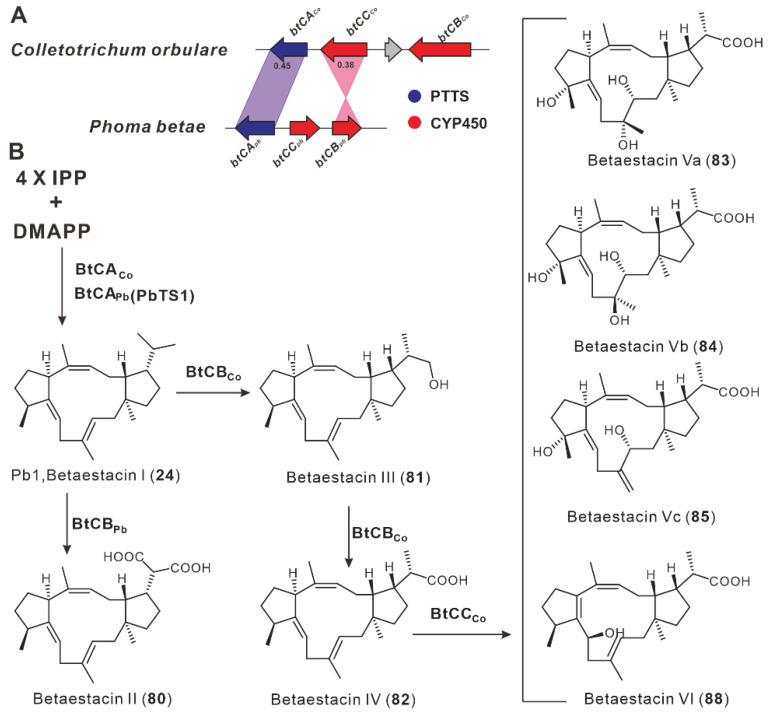
The BGC for betaestacins (**A**) and their biosynthetic pathway (**B**).

**Figure 10 jof-08-01080-f010:**
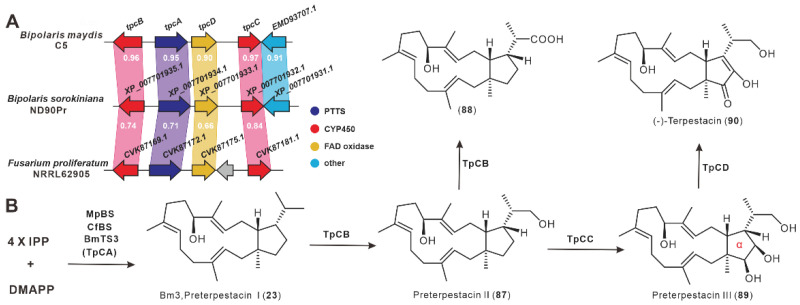
The *tpc* BGC (**A**) and their biosynthetic pathway (**B**).

**Figure 11 jof-08-01080-f011:**
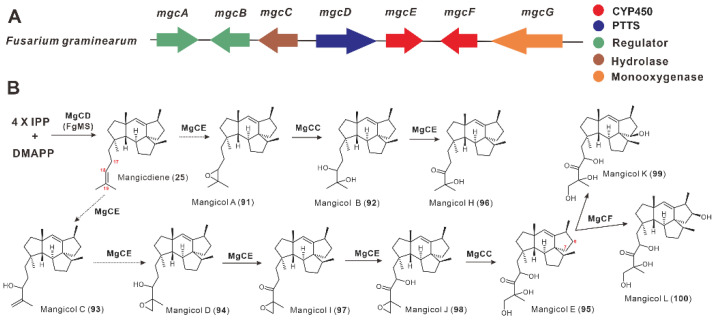
The *mgc* BGC (**A**) and mangicols biosynthetic pathway (**B**).

## Data Availability

Not applicable.

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
