# Peer review of "Research Progress on Fungal Sesterterpenoids Biosynthesis"

_jof, 2022, doi:10.3390/jof8101080_

Round 1
Author Response
Q1、I noticed that Authors named/nicked the enzymes involved in sesterterpenoids synthesis as PTTCs and in majors of cited publications the abbreviation of PTTC are used. I think that if Authors would maintained their abbreviation they have to explain why. Synthase and cyclase?
Chen, R.; Jia, Q.; Mu, X.; Hu, B.; Sun, X.; Deng, Z.; Chen, F.; Bian, G.; Liu, T. Systematic mining of fungal chimeric terpene 495 synthases using an efficient precursor-providing yeast chassis. Proc Natl Acad Sci U S A 2021, 118, e2023247118, 496 doi:10.1073/pnas.2023247118
A1. Thank you for your thoughtful advice. In order to maintain the rigour and legacy of academic research, we are happy to adopt PTTS as the abbreviation for sesterterpenoids synthesis, and have made adjustments in the manuscript.
Q2、Over the full text the fig. are in bold. Is it good?
A2. Thank you for your very kind suggestion. We have modified the font of "figure" and "table" throughout the text to match the format of the recent article in Journal of fungi.
A3、The abbreviations, although known but mentioned for the first time in the paper have to be full named. Line 26: “……precursors DMAPP and IPP.”
Q3, Thanks for your careful consideration, you pointed out a basic writing rule. In fact, the full names of DMAPP and IPP are given in the abstract (in lines 10-11).
Q 4、
(1)Line 38 Fungi-derived sesterterpenoids exhibit a diverse set.
(2)Line 52 action against Enterococcus faecalis and Pseudomonas aeruginosa [8]. .. Enterococcus faecalis and Pseudomonas aeruginosa.
(3)In table 1 Is Talaromyces Wortmannii has to be Talaromyces wortmannii Is Colletotrichum Orbulare has to be Colletotrichum orbulare in Wortmannii Orbulare.
(4)lines 268-269 Jiang et al. identified the fungal chimeric terpene synthases FoFS and AtAS from the plant pathogenic fungi F. oxysporum and A. terreus in 2021, respectively.
(5)In Fig 6 in the part C and E: Is Aspergillus Calidoustus has to be Aspergillus calidoustus Is Aspergillus Terreus has to be Aspergillus terreus.
(6)Lines 306-307 Then, the transporter gene oblDBm was co-expressed with ob lAAc and oblBBm lead to the three new sesterterpenes: ophiobiln C (71), 6-epioophiobiln C (72), and 6-epi- 307 ophiobolin N(73)[46] (Fig. 7B).
(7)In Fig 7: Is Aspergillus. clavatus has to be Aspergillus clavatus ( without .) Is Emericella. variecolor has to be Emericella variecolor ( without .)
(8)Line 329…77 continued to be further oxidised by another CYP450 oxidase, astA, to a new sesterterpe…
(9)In Fig 8 Is Aspergillus. clavatus has to be Aspergillus clavatus ( without .)
(10)Line 334 … estacin I (24) in two plant pathogenic fungi, P. betae and C. orbulare in 2018. Both BGCs…
(11)Line 335 … two CYP450… Line 337 …homologous CYP450s in these two BGCs. The first pair of CYP450s. Line 341 …. homologous CYP450s
(12)In Fig.9 Is Colletotrichum Orbulare has to be Colletotrichum orbulare Is Phona Betae has to be Phona betae
(13)Line 358 …production of 25 and was… ( bold for compound number)
(14)Lines 364 and 367 …multifunctional CYP450
(15)Lines 376-377 A total of 52 PTTCs were identified from 2013-2022, and most of them were functionally validated by heterologous expression using A. oryzae, E. coli or S. cerevisiae
A4. Thank you for the great effort you put into these details, and your scientific rigor is commendable. We are sorry for these omissions and have made appropriate corrections and modifications to all of them.
Q5、All over the “References” the names of fungi species (in the titles) are not in italic. Is it good? In position 49 ….Maruyama, J.-i.; …..?
A5. Thank you for your valuable advice. All species (Latin) names in this manuscript have been italicized.
6、I felicitate Authors for a clear presentation of complicated researches in this review paper.
A6. Thank you for your high recognition of our work, your affirmation inspires us.
Reviewer 2 Report
#### General comments:
This paper aims at complete coverage of the area while there is little
critique or synthesis which makes it a hard read. Nevertheless, this
compilation has merit on its own.
References to structures are given as bold numbers throughout the text.
This is good, but I request a table where the figure numbers are listed
so that the reader does not have to go through all figures to find the
structure. Better yet would be to renumber the structures so that they come
in the same order as the figures.
The English is fine throughout the paper.
#### Specific comments:
Line 38
...sesterterpen oidsexhibit...
Typo
Line 109
It is assumed that all known fungal PTTCs are Class I terpene synthases.
A reference is needed or rephrase and expand.
Line 182
Hong et al. conducted a systematic study on the biosynthesis of ophiobolin in A. ustus in 2016. Au8003 was confirmed to be the synthase responsible for ophiobolin F (15) in A. ustus by gene deletion and complementation experiments [25].
I assume that Hong et al. is reference 25? If so, add the reference to the first sentence as well.
Line 196
Hideaki Oikawa lab discovered four sesterterpene synthase genes from phytopathogenic fungi in 2017, including BmTS1, BmTS2, and BmTS3 from B. maydis, as well as PbTS1 from P. betae. The four sesterterpenes Bm1-3 (21-23) and Pb1 (24) were produced by heterologous expression of these four genes in A. oryzae and E. coli, respectively [30].
This is but one example of many where references are given to a laboratory, presumably run by Hideaki Oikawa. It would be more correct to reference the source as Narita et. al
I think this should be changed throughout the manuscript.
Author Response
Q1、This paper aims at complete coverage of the area while there is little critique or synthesis which makes it a hard read. Nevertheless, this compilation has merit on its own.
References to structures are given as bold numbers throughout the text.
This is good, but I request a table where the figure numbers are listed so that the reader does not have to go through all figures to find the structure. Better yet would be to renumber the structures so that they come in the same order as the figures.
The English is fine throughout the paper.
A1. Thank you for your approval of our manuscript as a whole and for your comments on the English language.
Your suggestion to summarize the compounds in a table and label them in the order of occurrence is something we have also considered. However, since this manuscript focuses on sesterterpene synthases, the description and discussion are centered on the enzymes, and the table summarizing the enzymes is also labeled with the relevant compounds.
In addition, the first 14 sesterterpene biosynthesis have not yet been identified and appear in the manuscript because of the need for a description of sesterterpene activity. sesterterpenoids are not the focus of this paper.
Since a list of sesterterpene synthases already exists in the manuscript, we have taken your suggestion in part to rearrange the compounds in the order of occurrence in the table about sesterterpene synthases to facilitate compound lookup.
Once again, thank you for your constructive comments, which will greatly help to improve the quality of our manuscript.
Q2、Line 38 ...sesterterpen oidsexhibit...Typo
A2. Thank you very much for your careful review, we have made the corrections that you suggested.
Q3、Line 109 It is assumed that all known fungal PTTCs are Class I terpene synthases.
A reference is needed or rephrase and expand.
A3. Thank you very much for your careful review. We have expanded on your suggestions and have cited a new reference.
Rudolf, J.D.; Chang, C.-Y. Terpene synthases in disguise: enzymology, structure, and opportunities of non-canonical terpene synthases. Natural Product Reports 2020, 37, 425-463, doi:10.1039/C9NP00051H.
Q4、Line 182. Hong et al. conducted a systematic study on the biosynthesis of ophiobolin in A. ustus in 2016. Au8003 was confirmed to be the synthase responsible for ophiobolin F (15) in A. ustus by gene deletion and complementation experiments [25]. I assume that Hong et al. is reference 25? If so, add the reference to the first sentence as well.
A4. Thank you very much for your careful review, and we have added citations as you suggested.
Q5、Line 196, Hideaki Oikawa lab discovered four sesterterpene synthase genes from phytopathogenic fungi in 2017, including BmTS1, BmTS2, and BmTS3 from B. maydis, as well as PbTS1 from P. betae. The four sesterterpenes Bm1-3 (21-23) and Pb1 (24) were produced by heterologous expression of these four genes in A. oryzae and E. coli, respectively [30]. This is but one example of many where references are given to a laboratory, presumably run by Hideaki Oikawa. It would be more correct to reference the source as Narita et. al, I think this should be changed throughout the manuscript.
A5. Thank you for your constructive comments, we have revised them accordingly.